# Neighborhood and Child Development at Age Five: A UK–US Comparison

**DOI:** 10.3390/ijerph181910435

**Published:** 2021-10-04

**Authors:** Anthony Buttaro, Ludovica Gambaro, Heather Joshi, Mary Clare Lennon

**Affiliations:** 1Independent Researcher, San Francisco, CA 94102, USA; abuttaro@gradcenter.cuny.edu; 2Department of Sociology, Eberhard Karls Universität Tübingen, 72074 Tübingen, Germany; 3Centre for Longitudinal Studies, Social Research Institute, University College London, London WC1H 0AL, UK; h.joshi@ucl.ac.uk; 4Graduate Center, The City University of New York, New York, NY 10016, USA; mlennon@gc.cuny.edu

**Keywords:** neighborhood effects, residential mobility, early childhood, cognitive development, externalizing problems, internalizing problems, longitudinal, comparative

## Abstract

Early childhood is a critical period in the life course, setting the foundation for future life. Early life contexts—neighborhoods and families—influence developmental outcomes, especially when children are exposed to economic and social disadvantage. Residential mobility, frequent among families with pre-school children, may reduce or increase exposure to adverse surroundings. We examine children’s cognitive and behavioral outcomes at age five, in relation to neighborhood composition, family circumstances and residential moves, using two longitudinal micro datasets: an urban subsample of the UK Millennium Cohort Study (N up to 7967), and the Fragile Families and Child Wellbeing Study in the US (N up to 1820). Each is linked to an index of neighborhood advantage, created to make UK/US comparisons, based on census and administrative information. A series of estimates indicate a strong association, in both countries, between cognitive scores and neighborhood advantage, attenuated but not eliminated by family circumstances. Children’s behavior problems, on the other hand, show less association with neighborhood advantage. There are minor and mixed differences by residential mobility particularly when neighborhood disadvantage changes. Notwithstanding the primacy of the family in predicting preschool development, the findings support the notion of neighborhood as potentially advantageous at least in relation to cognitive outcomes.

## 1. Introduction

In the last several decades, a large body of research from various disciplines has investigated the links between neighborhoods’ socio-economic conditions and individual well-being [1]. Spatial variations in well-being may be interpreted as ‘neighborhood effects’, but they may be confounded by area’s social composition. It is not clear whether the ‘effects’ of neighborhood disadvantage are sources or symptoms of individual disadvantage. This paper offers a contribution to research on spatial patterns in children’s development [2,3]. Two perspectives have guided this research. One from developmental psychology, rooted in Bronfenbrenner’ bio-ecological theory [4,5], conceptualizes human development as the product of the sustained interactions between children and their multiple nested environments. Children directly interact with more proximate contexts, such as family, daycare center, or neighborhood of residence, but are also indirectly affected by more distal environments, shaped by long-term economic, political, and cultural factors. The bio-ecological framework has been successful in emphasizing the embedded nature of children’s development, but empirical applications of this model have struggled to operationalize the complexity involved [2].

A second perspective that has guided research on neighborhood and children stems more directly from neighborhood research and highlights the neighborhood mechanisms and dimensions that are likely to be relevant to this age group [3,6]. Institutional resources catering for children, most notably childcare centers, recreational facilities, and medical facilities, have been found to be important channels linking neighborhood advantage and children’s achievement and well-being [7,8]. The indirect effects on children of stresses on parents’ housing [9,10], employment and health [11] have also been found to play a role in the experience of growing up in a disadvantaged neighborhood, as children at this age spend most of their time at home. Several non-experimental studies indicated that higher neighborhood disadvantage is associated with lower overall parenting quality, and these associations operate via heightened family stress, parents’ perception of dangers in their neighborhoods, and less social support [12,13,14,15]. However, there are also studies from the UK that failed to find evidence that parenting practices and characteristics of the family environment mediated the relationship between neighborhood disadvantage and children’s outcomes [11,16].

Findings from experimental mobility assistance studies such as the Gautreaux Program and Moving to Opportunity (MTO) have shown that relocation into more desirable neighborhoods can be beneficial for children’s education [17] and their mental health [18]. Longer term follow-up study of MTO by Chetty and colleagues [19] finds that children who move to low-poverty areas prior to age 13 have better educational and labor market outcomes. However, even experimental studies based on mobility assistance programs cannot fully answer developmental questions about life-course timing and the dosage of neighborhood effects [20].

In this paper we focus on children, and on the first five years of life in particular. Several reasons underpin researchers’ interest in children and neighborhoods. Children’s activities and interactions tend to be circumscribed by the area they live in, resulting in a strong conditioning of the social and environmental stimuli children encounter [21]. Early childhood is an especially sensitive developmental period, and alterations of a child’s ecology may have long-lasting effects [22]. Finally, mapping the relationship between neighborhood and children furthers our knowledge about people’s neighborhood histories and contributes to our understanding of neighborhood effects in life-course perspective, as recommended by van Ham and Manley [1].

We test whether neighborhood socio-economic composition at age one is associated with three developmental outcomes at age five. We contribute to the literature in two ways. First, we carefully account for residential mobility and thus for differential exposure to neighborhood. We include in our analyses moves leading to change in neighborhood socio-economic composition, distinguishing between trajectories entailing improvement or deterioration. This approach is especially important when investigating early childhood, a life stage characterized by high rates of residential mobility [23,24], which tends to accompany the family formation process [25,26].

Our second contribution is to take a comparative perspective and contrast results from two countries: the UK and the US. The literature on the developmental consequences of different neighborhood circumstances is predominantly from the US. In recent years, one research team has also conducted a number of studies on how UK children’s and adolescents’ development is associated with their area of residence [11,27,28,29]. Yet, it is not clear how comparable results from the two countries are, given the different measures, datasets and methods used. Comparing the UK to the US has the potential of bringing new insights to neighborhood research. At a broad level, the two countries share a common culture and economic system. Yet, relative to the US, the UK has a higher prevalence of relatively stable social housing tenancy [30], a less distinct pattern of residential segregation of ethnic minorities [31], and a more developed network of local services for families with children [32]. Although it remains difficult to directly investigate how neighborhoods may matter to children in two different macro contexts, the advantage of our approach is to ascertain whether the socio-economic composition of areas displays similar patterns of association in the two contexts.

## 2. Materials and Methods

### 2.1. Data and Comparative Analytical Approach

This paper uses data from two national birth cohort longitudinal studies: the Millennium Cohort Study (MCS) for the United Kingdom, and the Fragile Families and Child Wellbeing Study (FFCWS) for the United States. Both studies assessed child development, neighborhood, residential mobility, employment history, background characteristics, and health of both child and mother.

MCS is a nationally representative longitudinal study of children born in the UK between 2000 and 2002 [33]. MCS design over-sampled areas with high child poverty, high minority ethnicity (in England only), and the three smaller countries of the UK. Altogether 19,243 families have been interviewed and, as we explain later, we select here more than 10,500 children born in large cities. The first interviews with the main caregiver (almost always the biological mother) were conducted when children were 9 months old followed by face-to-face surveys when they were 3, 5, 7, 11, 14 and 17 years old, so far [34,35,36,37].

FFCWS is a panel study of 4898 families with children born between 1998 and 2000 selected in 20 large US cities with an oversampling of unmarried parents [38]. FFCWS is used widely to investigate disadvantaged families in cities [39], which is the population for whom previous research has reported higher residential mobility [40]. Extensive information was collected starting from the birth of the focal child and subsequently at 1, 3, 5, 9, and 15 years of age, so far. When children were age 5 (in the fourth round of data collection), 74% of mothers participated in an additional in-home survey module, through which the cognitive skills of 57% of children were assessed.

We selected two waves of data collected when children in both surveys were about the same age, namely: 9 months and 5 years in MCS; 1 and 5 years in FFCWS (in the rest of the paper we use ‘1 year’ or ‘1 year wave’ for both ages, 9 months in MCS, and 1 year in FFCWS). All interviews were conducted from 2001 through 2007 in the UK, and from 1999 through 2006 in the US. The temporal window in which we studied neighborhood patterns in child’s outcomes is approximately four years long (1–5 years).

To increase the comparability of the datasets, we selected children from the MCS who were born in hospitals in large cities, defined as having population above 100,000 (detailed information on the selection procedure is available upon request). Our comparison between the two countries is thus confined to children born in urban areas, although hereinafter we refer to the two samples as UK and US, respectively. To further increase comparability: the analytic samples were restricted to survey respondents who were the biological mother. Moreover, only children with complete data on residential mobility, neighborhood composition, and outcome variables were included. For MCS these selections resulted in analytic sample of 7967 city-born cohort members with data on verbal skills, and 7688 with behavior scores; for FFCWS, the respective sample sizes were 1458 and 1820. Comparison of the characteristics of full and analytic samples showed in general very small differences. The analytic sample in MCS had an underrepresentation both of mothers born abroad and in the White ethnic group (respectively −3.4% and −2.7%). For the US sample there was an under-representation of mothers born abroad (−3.3%) and an over representation of the Black ethnic group (+2.0%) in the verbal sample.

Missing data on covariates were imputed using a two-step procedure. First, we logically replaced some of the missing data using a version of the variable from other waves. Secondly, remaining missing data were imputed via Rubin’s Multiple Imputation (MI) procedure [41]. In our multivariate models, parameters of interest were estimated and averaged across 20 data sets, and then adjusted for missing data uncertainty. In all analyses we controlled for survey design effects (i.e., probability weights, sampling strata, and primary sampling units), using the *svy* command in Stata version 14.

### 2.2. Variables

Variables were chosen and operationalized to maximize comparability across the two samples [42]. Descriptive statistics are reported in Table 1; Appendix A presents additional univariate statistics and correlation matrices of main and additional variables.

#### 2.2.1. Dependent Variables

Three outcomes of child development at age five were used in the analyses: Verbal score at 5 years (percentile), Externalizing adjustment at 5 years (percentile), and Internalizing adjustment at 5 years (percentile). The behavior difficulty scores were inverted so that all development scores moved in a positive direction.

The original comparative measures of verbal ability were: for the UK sample, the Naming Vocabulary from the British Ability Scales II (BAS II) [43]; for the US sample, the Peabody Picture Vocabulary Test (PPVT) [44]. Although they assess expressive language skills and receptive vocabulary, respectively, they are each designed to capture children’s linguistic skills through identification of everyday objects on test show-cards and have been used before in comparative research [45]. They are age-normed tests based on the national population of 5-year-old children with Mean = 50 and SD = 10, in UK, and Mean = 100 and SD = 15, in US. Both analytic samples show means and standard deviations that depart from the national norms.

The original Externalizing and Internalizing behavior scales capture children’s behavioral and emotional problems. They consisted of mothers’ responses to items from the Strengths and Difficulties Questionnaire (SDQ) [46] in the UK, and the Child Behavior Checklist (CBCL) [47] in the US. The SDQ Externalizing measure consisted of the sum of 10 items from the ‘Conduct problem’ (e.g., often has temper tantrums, fights/bullies other children) and ‘Hyperactivity’ sub-scales (e.g., easily distracted, fidgeting); the Internalizing outcome summed a total of 10 items belonging to the ‘Emotional’ (e.g., often worried, unhappy) and ‘Peer problems’ sub-scales (e.g., tends to play alone, bullied by other children). Cronbach’s alpha for the externalizing and internalizing measures were, respectively 0.66 and 0.79. The CBCL Externalizing scale assessed acting-out forms of behavioral problems (e.g., argues a lot, disobedient at home/school, lies/cheats) (mean score of 30 items; alpha = 0.86). The Internalizing scale covers emotional problems (mean score of 22 items; Cronbach’s alpha = 0.75).

To facilitate comparisons between the two countries, we transformed all three outcomes into percentile scores ranging from 1 for children with the lowest scores, to 100 for the highest scoring children. This approach has been used in educational and comparative research [48,49]. It increases stability by reducing the influence of extreme values without increasing *t*-test Type-I error across a wide variety of distributions [50]. When averaged across outcomes, the correlation between original scores and corresponding percentile version was 0.94 in the UK, and 0.95 in the US. In Table 1 we show that the bivariate correlation between each dependent variable and the neighborhood measure is not affected at all by the above transformations. We further confirmed the stability of results in the multivariate analysis stage across original and percentile version of the same outcome (results available upon request).

#### 2.2.2. Independent Variables

We present the predictors used in the multivariate analysis according to their theoretical domain, that is: area and mobility, family context, health, and basic control variables.

Area and mobility. *Area social advantage at 1 year* (percentile) is the measure of quality of the neighborhoods where family lived at the age 1. It is a composite score obtained by Principal Component Analysis (PCA) run on a series of census-based indicators of socio-economic conditions of residential areas [51]. To enhance cross-country comparability (within UK, and across UK–US), we chose the ecological unit(s) of analysis in terms of population size. In the UK, this meant selecting Lower-level Super Output Area (LSOA) for both England and Wales, Data-Zone (DZ) for Scotland, and Electoral/Council Ward for Northern Ireland, resulting in an average area (i.e., LSOA/DZ/Ward) population Mean of 1418 and SD of 238. In the US, the corresponding ecological unit was the census tract with Mean = 4300 and SD = 2142. As described in detail by Buttaro and Gambaro [51], we took measures of disadvantage/advantage from the 2000 US Census and, for the UK, from the 2001 Census and administrative data. Six measures common to both countries were the proportions of: unemployed in the labor force; households receiving welfare; households headed by females; adults with no high school diploma or UK equivalent; adults with a college degree; and adults who are managerial/professional workers; a seventh variable selected for the US index was proportion of people below poverty level not available for small areas in the UK. We created the index at the national level first, and then merged to both data sets by the geocodes of surveyed addresses at 1 year and 5 years. Higher scores on the resulting index indicates greater socio-economic advantage at the area level, furthermore, we transformed the index at both waves on a percentile scale.

We computed the *change in area social advantage 1–5 years* (percentile differences) as the difference between the 5 year and the 1 year percentile measures of the area social advantage. The range of this variable runs from negative through positive, and the ‘0’ scores refer to both children who did not change residence, and those who moved within/to an area of equal rank on the index. A negative score on this index indicates that a residential move produced a decline in neighborhood advantage (i.e., area score higher at age 1 than age 5); a positive score means that moves resulted in greater neighborhood advantage (i.e., area score lower at age 1 than age 5). There is no allowance for neighborhood change being experienced by those who do not move, as the social advantage of each area is only assessed once. We assume that changes in neighborhood composition only occur relatively slowly and that it is not unreasonable to assume the neighborhood advantage remains stable over a period of four years [52].

*Moved between 1–5 years* is a binary measure (‘1 = yes’) that captures whether the family of the focal child had moved between 1 year and 5 year rounds. This was based on respondents’ answers to the question of whether their address was the same as at previous interviews. We opted for a dichotomous variable instead of a move count because of evidence that the number and timing of family moves added little to the explanation of variations in child outcomes [53].

Family Context. *Household was workless at 1 year* indicated whether, at the 1 year wave, in cohort member’s household at least one parent (or domestic partner) was employed/doing paid work (coded ‘0’), or neither was employed (coded ‘1’). *Mother was single at 1 year* was coded ‘0’ if mother was partnered (without distinction between married and cohabiting couples), and ‘1’ if single. *(LN) Equivalized income at 1 year* was the reported household income, adjusted according to household size and composition. To reduce the influence of outliers, in the analyses we used the (natural) logarithmic transformation of this measure. *Mother’s level of education* was created to approximate content-equivalence in the different educational systems in UK and US. It ranged from ‘1 = minimal formal qualification (UK)/less than 9th grade (US)’ through ‘7 = higher degree/graduate degree’. *Race/Ethnicity* refers to the mother’s ethnic background and it is a five-category measure in the MCS sample (i.e., White, Black, Indian, Pakistani/Bangladeshi, and Other race/ethnicity), and a four-category variable in FFCWS (i.e., White, Black, Hispanic, and Other race/ethnicity). We note that groups with the same label (i.e., ‘White’ and ‘Black’) have different social origins and histories of spatial segregation in the two countries. *Mother was not born in UK/US* is a dummy variable (‘1 = yes’) indicating the immigrant status of focal child’s mother. Household size at 1 year refers to the number of people living in the household including the respondent. *Mother’s age* in years at cohort member’s birth records her age when focal child was born. *Cohort member was the first child* is a binary variable coded ‘l = yes’ when the focal child did not have any older sibling at birth. *Housing tenure* at 1 year is a five-category measure: Public housing, Subsidized rented housing, Market rented housing, Owned housing, and Shared/other types of housing.

Health. *Cohort Member was born underweight* is ‘1 = less than 2500 gm’ and ‘0 = 2500 gm or more’. *Cohort Member’s general health at 3 years/1 year* refers to the mother’s assessment of focal child’s health conditions. In MCS, the measure was collected at the 3 year wave as a dichotomy when mothers were asked whether the child had “had any longstanding health condition”, coded here as ‘1 = yes’. In FFCWS it was collected at age 1 with responses ranging from ‘1 = poor’ through ‘5 = excellent’. Despite the different timing, direction and scales, the measures represent similar concepts. *Mother’s general health at 1 year* wave is based on self-reported health at the first wave but the studies had different ranges: in MCS ‘1 = up to fair’ through ‘3 = excellent’; in FFCWS it was ‘1 = poor’ through ‘5 = great’. *Mother depressed at 1 year* wave is a binary variable (‘1 = yes’) measured in MCS by the 1 year survey item asking the mother whether she had ever been diagnosed by a doctor with depression or serious anxiety; in FFCWS, it is based on meeting criteria for a major depressive episode (MDE) on the composite International Diagnostic Interview Short Form [54].

In supplemental analyses we used six measures of changes that may have occurred across surveys within families, by combining 1 year and 5 year measures for employment and family status, income, and household size. These were added to examine whether change in these variables affected our estimates of neighborhood associations. Results, shown in Appendix A, Tables (Model 4) suggest that these change measures do not alter our substantive findings.

Basic control variables. Cohort Member’s sex is male is coded ‘1’ when the focal child is a boy. Cohort Member’s age in months at 5 years controls for fieldwork variability in the age at the time of interview.

### 2.3. Analytical Strategy

The main hypothesis of the study is about the association of neighborhood with both cognitive development and behavioral adjustment of children. We predicted each of the three outcomes using Ordinary Least Square (OLS) regressions, adding the variables in three sequential models. All models included the basic control variables (sex of child and age). Then: Model 1 tested the associations between neighborhood advantage at 1 year and child outcomes at 5 years (adjusted for child sex and age). Model 2 allowed for the change in neighborhood advantage that occurred as consequence of residential mobility within the 1–5 year time window. Model 3 investigated how these associations changed when the family context and health at baseline year were taken into account. This domain comprised family vulnerabilities and capabilities, such as household employment and mother’s partnership status, income, background characteristics of both households (e.g., size and tenancy) and mothers (e.g., education, ethnicity immigrant status, first child, etc.), as well as health conditions of both cohort member and mother. Each outcome was explored in turn.

As reported above, we opted for a percentile transformation of the neighborhood measures (i.e., 1 year and change 1–5 years) to increase comparability of results across samples. One concern was that such transformation could introduce bias in the estimation, ultimately affecting the interpretation of the results. Therefore, we ran a two-step parallel analysis to investigate the sensitivity of the results to different “versions” of both variables: in the first step, we ran the regression models with our percentile transformation against equivalent models using other metrics (i.e., original factor scores, squared, natural logarithmic, various quantile cuts), obtaining interchangeable results. In the second step, having selected the percentile transformation, we tested it against one replacing the change in neighborhood advantage 1–5 years with a version split into two variables—negative change (with positive values = 0), and positive change (with negative values = 0). These analyses produced slightly different results in the UK and US samples for verbal scores, discussed after the main results in the next section.

## 3. Results

Table 1 reports the descriptive statistics of the variables used in the analysis, including the correlation with the area social advantage score (in percentiles) at the 1 year wave. In both samples, children appeared to have better outcomes when living in areas with higher advantage scores. The correlation with verbal scores was stronger than with socio-emotional adjustment, especially in the US sample.

Patterns of residential mobility differed between the two sample. Whereas in the UK sample 40 percent of children moved between the 1 year and 5 year waves, in the US sample 63 percent did. Yet, despite lower mobility, the average change in area social advantage was larger in the UK than in the US. Housing tenure also differed starkly between samples, with a much stronger prevalence of owner occupation in the UK sample than in the US one. Additionally, public housing was far more common in the UK (22%) than in the US (7.8%).

### 3.1. Area and Children’s Verbal Scores

Table 2 reports the three OLS regression models used to estimate the association between children’s verbal scores and area social advantage. All models control for cohort member’s sex and exact age at assessment. The coefficient of area social advantage shows that, in both sample, children who were living in more advantaged areas at age 1 also displayed higher verbal scores at age 5. Model 2 includes an indicator of whether the child moved and a measure of any ensuing change in area social advantage. While residential mobility was not associated with verbal outcomes in either sample, area change was. Improvements (deteriorations) in area social advantage were associated with higher (lower) verbal scores in the 5 year wave. This indicated that both the area from which the children were starting and the direction of area change mattered to the verbal scores in both samples, and particularly so for those from the US. Model 3 controlled for family context and child and maternal health. Their inclusion substantially reduced the magnitude and statistical significance of the parameters of area advantage and change. There were however some interesting differences in results between the two samples. In the UK sample, the area social advantage at age 1 continued to display a (reduced) positive coefficient but the change in area advantage did not. The reverse occurred in the US sample: improvement or deterioration in area advantage remained associated with verbal scores, irrespective of the level of advantage of the initial area. The association with area score at age 1 in the US was effectively accounted for by the individual circumstances included in Model 3. In both countries the level of family income was a significant predictor of verbal outcomes (Appendix A). The most powerful contribution to accounting for area differences were ethnic/ immigrant group and housing tenure, which are particularly strongly associated with the area score at age 1 in the US sample (Table 1). The predictors of child verbal scores which are significant in the UK sample are age of mother, household size, birth order, mother’s education and child health (Appendix A). However, they are not, apart from the young motherhood, more strongly associated with neighborhood score in UK than US (Appendix A). In neither sample was the correlation between area social advantage at age 1—either expressed in percentile or as original score—and residential mobility statistically or substantively significant, indicating that there was no apparent difference between the two samples in the selection into residential mobility. Thus, the greater spatial patterning of verbal scores in the US reflects greater social patterning in mobility and segregation in the family context, particularly those involving race and housing tenure.

Additional analyses, taking the direction of neighborhood change into account, are shown in Appendix A. We substituted change in neighborhood advantage 1–5 years with two distinct variables—negative change (with positive change set to 0), and positive change (with negative change set to 0). In the UK sample, only positive neighborhood change showed a statistically significant, albeit very small, positive association with verbal scores; in the US sample, negative neighborhood change was associated with worse verbal scores. This suggests that in the US children’s verbal skills were more sensitive and likely to be lower among children who had experienced a disadvantageous neighborhood change. All in all, the UK sample appeared characterized not only by a lower prevalence of residential mobility, but also by a greater persistence of the association between the social advantage of the area where children lived at age 1 and their verbal scores four years later. In the US sample, the change in area advantage that families experienced—especially if it was a negative change—mattered more to children’s verbal abilities.

### 3.2. Area and Children’s Externalizing Adjustment

We use the same three models to investigate the patterns of association between area social advantage and children’s externalizing adjustments, measured as low levels of externalizing problems (Table 3). Compared to verbal scores, the overall level of fit was lower and the associations weaker, particularly for the US. In Model 1 the estimates indicate that in both samples, children living at age 1 in areas with higher advantage score had better externalizing adjustment. Accounting for residential mobility in Model 2 did not substantially change these coefficients. However, estimates from the UK sample suggest that children who had moved home had better externalizing adjustment than their residentially stable peers, and changes in area social advantage further showed a positive, albeit very small, association. With the inclusion of controls for family context and measures of maternal and child health, the area coefficients in the UK sample became negligible in magnitude and mostly statistically insignificant. In contrast to the analysis of verbal scores, neither ethnic group nor income were significant in Model 3. The variables helping to account for spatial differences in the UK Model 3 were public housing tenure, mother’s age, education and health of mother and child. The first three of these have a strong spatial pattern. By contrast, in the US sample, almost the only significant coefficient was on residential mobility indicating a large positive association with externalizing adjustment (the only other significant individual estimate was on maternal heath in Model 3). Change of area, irrespective of magnitude of the change experienced, appeared to matter more among children from the US sample than in the UK.

### 3.3. Area and Children’s Internalizing Adjustment

Table 4 reports results on internalizing adjustment. Here, again the initial association was positive, indicating that children who, at age 1, were living in more advantaged areas displayed greater internalizing adjustment than their peers who had lived in less advantaged areas, particularly in UK. The inclusion of residential mobility and of change in area social advantage did not substantially alter these associations. However, in contrast to results on externalizing adjustment, children in the UK sample displayed more internalizing problems if they had moved. Neither residential mobility nor change in area social advantage reached statistical significance in the US sample. With the inclusion of family and health controls in Model 3, all area coefficients in both samples lost significance, except for area social advantage at age 1 in the UK sample, which remained tenuously associated with better internalizing adjustment. In both countries, the variables that contributed to the explanation of Model 3 were ethnic group and mother’s general health. In the UK sample there were also significant contributions from household income, mother’s age, mother’s education, maternal depression, child’s birth order and child health. Overall, then internalizing behavior did not appear to be particularly sensitive to area advantage, except through family context and health.

## 4. Discussion

This study is one of the first, to our knowledge, to compare the associations between children’s early development and neighborhood socio-economic composition in two countries—the UK and the US. We examined three developmental outcomes—verbal skills, externalizing adjustment, and internalizing adjustment—measured at age five, when children have just started or are about to start compulsory schooling. Our focus on early childhood is motivated by the foundational role of this life period [55] and its hypothesized high sensitivity to area influences [21]. While young children’s daily activities tend to take place within the neighborhood, early childhood is also a life phase characterized by high residential mobility [23,24,56]. A further contribution of this study is that it complements its examination of neighborhood socio-economic composition at one point in time with a consideration of neighborhood change to account for residential mobility.

With the advantage of two large birth cohort studies and after extensive work to make the analytical samples and measures used comparable, we found broadly similar results in the two countries. In particular, the cognitive outcome appeared to be more susceptible to neighborhood variation than the behavioral scores. Associations of initial area advantage or it change were detectable with verbal skills after accounting for a wide range of family characteristics, whereas socio-emotional outcomes were less so, particularly in the US sample. There remained however some noticeable differences between the two samples. In the UK sample, children’s verbal skills were associated with moving to a more advantaged area, while in the US they appeared to be more vulnerable to moves to a less advantaged area. This finding partly diverges from previous evidence that cognitive outcomes were sensitive to positive area changes, and behavioral ones were more closely associated with negative area changes [3]. Our cross-country comparative approach, although not able to identify causal mechanisms, proved helpful in highlighting the vulnerability, in terms of verbal development, to disadvantageous moves of children in US cities. We can also suggest that the greater spatial variation in verbal scores for the US children is associated with the higher degree of racial segregation and less secure housing tenure in US cities. Vulnerability of verbal and internalizing scores to insecure housing tenure is also apparent in the UK results. The findings from US may be indicative of what might be expected in UK as the housing available to young families in the UK becomes more insecure in more recent years. In both countries the health variables in Model 3 help to account for variations in child development while not displaying a strong spatial pattern, especially in the behavior outcomes. It should also be noted that results vary in detail by dependent variable as well as by country, so we urge caution about drawing generalized conclusions.

This study has a number of limitations. As is common in the quantitative literature on neighborhood, we relied on statistical geographical areas that may not correspond either conceptually or empirically with the boundaries relevant to children and their families [57,58,59]. The size of the areas differed between the two country samples, with larger units in the US. Despite this asymmetry, census tracts in the US appeared to display greater homogeneity than areas in the UK [51]. Our measure of neighborhood advantage captures the socio-economic profile of the resident population but does not reflect other neighborhood characteristics or processes that have been shown to matter for children, such as availability of green areas [28,29], exposure to environmental pollutants [60,61] or social networks [2]. It also does not capture changes in ecological conditions over time, which may be important in the longer term.

Another limitation is that the two cohort studies examined—the MCS and the FFCWS—although broadly similar, were not designed with comparison in mind and do not employ exactly the same outcome measures. The smaller sample size of the FFCWS means that it is not as highly powered as the MCS, resulting in less precise estimates. Nevertheless, both sources tap very similar verbal abilities and behavioral outcomes allowing us to test whether patterns of associations are invariant to macro-context. Based on two samples of children born in large cities, our findings are not generalizable to the whole population. It is also important to consider the historical times examined here, as both surveys predate the 2008 recession. Changes in the housing market and to local services since then are likely to have transformed neighborhoods and resulted in different geographies of advantage and disadvantage.

## 5. Conclusions

This study has shown that there is a residual association between children’s verbal skills at age five and the socio-economic composition of the neighborhood(s) they have lived during early childhood. These associations were detectable in both the UK and US samples, supporting the hypothesis that childhood is a sensitive life stage to neighborhood effects as children are particularly dependent on local services and networks. Although further research is needed to establish whether these associations are causal and what mechanisms underpin them, policy initiatives aimed at improving family support services in disadvantaged areas hold the potential of levelling the playing fields for younger generations.

## Figures and Tables

**Table 1 ijerph-18-10435-t001:** Weighted Descriptive Statistics in UK and US.

Variables by Domain	UK					US				
	N	Mean	SD	Range	Corr. Area Soc. Adv. 1 yr	N	Mean	SD	Range	Corr. Area Soc. Adv. 1 yr
Dependent variables										
Verbal score at 5 yrs (original)	7967	54.6	11.1	20–80	0.31 ***	1458	95.2	17.6	40–133	0.43 ***
Verbal score at 5 yrs (percentile)	7967	52.7	28.0	1–99	0.32 ***	1458	53.8	30.5	1–100	0.44 ***
Externalizing behavior at 5 yrs (original)	7583	4.6	3.3	0–19	−0.22 ***	1820	0.4	0.2	0.0–1.5	−0.17 ***
Externalizing behavior adjustment at 5 yrs (percentile)	7583	56.7	29.3	1–100	0.21 ***	1820	53.5	28.6	1–100	0.16 ***
Internalizing behavior at 5 yrs (original)	7606	2.4	2.5	0–18	−0.20 ***	1820	0.2	0.2	0.0–1.1	−0.17 ***
Internalizing behavior adjustment at 5 yrs (percentile)	7606	60.5	30.9	1–100	0.21 ***	1820	53.7	30.7	1–100	0.16 ***
Area and mobility										
Area social advantage at 1 yr (percentile)	7967	50.5	28.8	1–100	—	1820	49.2	29.0	1–100	—
Area social advantage at 5 yrs (percentile)	7967	50.5	28.8	1–100	0.83 ***	1820	48.7	29.4	1–100	0.78 ***
Moved between 1–5 yrs	7967	0.399	—	0–1	−0.02 *	1820	0.628	—	0–1	−0.02 *
Change in area social advantage 1–5 yrs (diff. of percentiles)	7967	0.7	15.4	−91–95	−0.27 ***	1820	0.0	19.6	−76–92	−0.31 ***
Family context										
Household was workless at 1 yr	7478	0.159	—	0–1	−0.35 ***	1804	0.139	—	0–1	−0.34 ***
Mother was single at 1 yr	7967	0.135	—	0–1	−0.27 ***	1811	0.197	—	0–1	−0.30 ***
LN equivalized income at 1 yr	7875	9.7	0.7	6.8–11.2	0.56 ***	1820	9.9	1.4	0.0–13.0	0.54 ***
Mother’s level of education	7950	4.0	1.7	1–7	0.48 ***	1820	4.0	1.8	1–7	0.52 ***
Race/Ethnicity										
White (ref. group)	7967	0.855	—	0–1	0.15 ***	1820	0.383	—	0–1	0.52 ***
Black	7967	0.036	—	0–1	−0.07 ***	1820	0.246	—	0–1	−0.41 ***
Hispanic	—	—	—	—	—	1820	0.300	—	0–1	−0.26 ***
Indian	7967	0.028	—	0–1	−0.01 ***	—	—	—	—	—
Pakistani/Bangladeshi	7967	0.050	—	0–1	−0.17	—	—	—	—	—
Other race/ethnicity	7967	0.030	—	0–1	−0.01 ***	1820	0.071	—	0–1	0.19 ***
Mother was not born in UK/US	7089	0.122	—	0–1	−0.03 **	1816	0.208	—	0–1	−0.14 ***
Household size at 1 yr	7967	4.0	1.2	2–12	−0.11 ***	1814	4.4	1.5	1–15	−0.26 ***
Mother’s age in years at cohort member’s birth	7966	29.3	5.8	14–49	0.38 ***	1820	26.8	6.2	14–47	0.29 ***
Cohort member was first child	7967	0.447	—	0–1	0.07 ***	1819	0.364	—	0–1	0.14 ***
Housing Tenure at 1 yr										
Public housing	7952	0.222	—	0–1	−0.45 ***	1818	0.078	—	0–1	−0.31 ***
Subsidized rented housing	7952	0.025	—	0–1	−0.07 ***	1818	0.051	—	0–1	−0.16 ***
Market rented housing	7952	0.045	—	0–1	0.03 *	1818	0.325	—	0–1	0.01
Owned housing (ref. group)	7952	0.652	—	0–1	0.44 ***	1818	0.336	—	0–1	0.39 ***
Shared/other types of housing	7952	0.056	—	0–1	−0.08 ***	1818	0.210	—	0–1	−0.15 ***
Health										
Cohort Member was born underweight	7959	0.073	—	0–1	−0.03 **	1775	0.061	—	0–1	−0.08 **
Cohort Member’s general health at 3 yrs/1 yr	7089	0.159	—	0–1	−0.04 ***	1818	4.5	0.8	1–5	0.11 ***
Mother’s general health at 1 yr	7963	2.2	0.7	1–3	0.19 ***	1820	3.7	1.0	1–5	0.15 ***
Mother depressed at 1 yr	7963	0.229	—	0–1	−0.10 ***	1820	0.122	—	0–1	−0.03
Changes in family context 1–5 yrs										
Household was workless at 5 yrs	7476	0.047	—	0–1	−0.11 ***	1802	0.092	—	0–1	−0.06 ***
Household was employed at 5 yrs	7476	0.064	—	0–1	−0.17 ***	1802	0.089	—	0–1	−0.28 ***
Mother was single at 5 yrs	7966	0.086	—	0–1	−0.09 ***	1811	0.171	—	0–1	−0.04
Mother was coupled at 5 yrs	7966	0.047	—	0–1	−0.15 ***	1811	0.074	—	0–1	−0.19 ***
Change in LN equivalized income 1–5 yrs	7802	0.1	0.5	−3.8–3.4	−0.09 ***	1820	0.1	1.1	−10.9–10.5	−0.09 **
Change in household size 1–5 yrs	7966	0.3	1.1	−9–9	0.08 ***	1803	0.1	1.5	−8–8	0.15 ***
Basic control variables										
Cohort Member’s sex is male	7967	0.506	—	0–1	0.00	1820	0.559	—	0–1	−0.02
Cohort Member’s age in months at 5 yrs	7967	62.6	2.9	53–74	0.04 ***	1802	60.9	2.2	56–71	0.08 **

* *p* < 0.05; ** *p* < 0.01; *** *p* < 0.001. “yr” and “yrs” stand for “year” and “years” and refer to the age of the cohort member at the survey wave when the relevant variable was collected.

**Table 2 ijerph-18-10435-t002:** Verbal Score at 5 years: Selected OLS Unstandardized Regressions Coefficients (Standard Error in parentheses) in UK (N = 7967) and US (N = 1458) ^a,b^.

Variable	Model 1: Area Social Advantage		Model 2: Area and Mobility		Model 3: Model 2 + Family Context & Health	
	UK	US	UK	US	UK	US
Area social advantage at 1 year (percentile)	0.31 *** (0.02)	0.47 *** (0.09)	0.33 *** (0.02)	0.58 *** (0.06)	0.11 *** (0.02)	0.10 (0.11)
Change in area social advantage 1–5 years (diff. of p-tiles)	—	—	0.16 *** (0.02)	0.47 ** (0.11)	0.04 (0.02)	0.23 ** (0.08)
Moved between 1–5 years	—	—	−0.40 (0.87)	2.55 (5.53)	−0.91 (0.82)	1.99 (3.29)
Constant	59.41 *** (7.89)	48.15 (60.05)	59.49 *** (7.89)	10.22 (55.92)	25.81 * (10.75)	31.15 (41.07)
*F*-test	104 ***	70 ***	70 ***	57 ***	133 ***	10,189 ***
Adjusted *R*^2^	0.10	0.20	0.11	0.28	0.24	0.46

* *p* < 0.05; ** *p*< 0.01; *** *p* < 0.001. ^a^ All models include Cohort Member’s sex and age at 5 year wave. ^b^ Full results for Model 3 are reported in Appendix A.

**Table 3 ijerph-18-10435-t003:** Externalizing Behavior Adjustment at 5 years: Selected OLS Unstandardized Regressions Coefficients (Standard Error in parentheses) in UK (N = 7668) and US (N = 1820) ^a,b^.

	Model 1: Area Social Advantage		Model 2: Area and Mobility		Model 3: Model 2 + Family Context & Health	
	UK	US	UK	US	UK	US
Area social advantage at 1 year (percentile)	0.22 *** (0.01)	0.15 ** (0.05)	0.23 *** (0.01)	0.17 ** (0.05)	0.04 * (0.02)	0.10 (0.10)
Change in area social advantage 1–5 years (diff. of p-tiles)	—	—	0.07 ** (0.03)	0.07 (0.06)	−0.03 (0.03)	0.02 (0.05)
Moved between 1–5 years	—	—	2.46 ** (0.77)	3.71 (2.61)	−0.72 (0.78)	5.01 ** (1.51)
Constant	32.72 *** (8.48)	74.18 ** (23.21)	32.48 *** (8.44)	72.97 *** (22.40)	9.43 (11.26)	46.45 (23.53)
*F*-test	168 ***	6	103 ***	4 *	52 ***	810 ***
Adjusted *R*^2^	0.06	0.03	0.07	0.03	0.13	0.09

* *p* < 0.05; ** *p* < 0.01; *** *p* < 0.001. ^a^ All models include Cohort Member’s sex and age at 5 year wave. ^b^ Full results for Model 3 are reported in Appendix A.

**Table 4 ijerph-18-10435-t004:** Internalizing Behavior Adjustment at 5 years: Selected OLS Unstandardized Regressions Coefficients (Standard Error in parentheses) in UK (N = 7668) and US (N = 1820) ^a,b^.

	Model 1: Area Social Advantage		Model 2: Area and Mobility		Model 3: Model 2 + Family Context & Health	
	UK	US	UK	US	UK	US
Area social advantage at 1 year (percentile)	0.22 *** (0.01)	0.16 ** (0.05)	0.24 *** (0.02)	0.19 * (0.07)	0.07 ** (0.02)	0.15 (0.12)
Change in area social advantage 1–5 years (diff. of p-tiles)	—	—	0.12 *** (0.03)	0.15 (0.11)	0.03 (0.03)	0.13 (0.11)
Moved between 1–5 years	—	—	−1.54 * (0.75)	−0.32 (3.40)	−0.51 (0.73)	0.59 (3.59)
Constant	42.04 *** (9.56)	152.90 ** (45.57)	41.96 *** (9.59)	142.93 ** (45.57)	14.88 (13.75)	154.37 *** (32.21)
*F*-test	74 ***	12 ***	45 ***	12 ***	42 ***	2571 ***
Adjusted *R*^2^	0.04	0.04	0.05	0.05	0.10	0.11

* *p* < 0.05; ** *p* < 0.01; *** *p* < 0.001. ^a^ All models include Cohort Member’s sex and age at 5 year wave. ^b^ Full results for Model 3 are reported in Appendix A.

## Data Availability

The main data reported in this manuscript were obtained from publicly available data. The Fragile Families and Child Wellbeing Study (FFCWS) interview data are available through Princeton University’s Office of Population Research (OPR) data archive, see https://fragilefamilies.princeton.edu/documentation (accessed on 28 September 2021). FFCWS data including geographic identifiers are available upon application at https://fragilefamilies.princeton.edu/restricted (accessed on 28 September 2021). Most of UK Millennium Cohort Study (MCS) data are available through the UK Data Service, at https://beta.ukdataservice.ac.uk/datacatalogue/series/series?id=2000031 (accessed on 28 September 2021). Some MCS data used in this study are not publicly available and were provided to the authors by the study owner: Centre for Longitudinal Studies, University of London. The analysis code used in this study is available by emailing the corresponding author.

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
