# Peer review of "Neighborhood and Child Development at Age Five: A UK–US Comparison"

_ijerph, 2021, doi:10.3390/ijerph181910435_

Round 1
Reviewer 1 Report
This is a well-executed study and well-written paper on an important and timely topic. I had just a few suggestions to improve the manuscript:
- In discussing how/why neighborhoods might affect young children, I thought the authors could say more about how the most important mechanisms are likely to be indirect - i.e. via effects on parents' health, mental health, stress, etc. -- because very young children are mainly at home with their parents.
- The authors drop cases that are missing the key outcomes and predictors. They mention that there are generally small differences between the full sample and the analytic sample but they should provide more detail on this point.
- The manuscript needs to be very carefully proofread and edited. I spotted numerous typos and errors - e.g. child' instead of child's, yrs years instead of yrs or years, etc.
Reviewer 2 Report
Neighborhood and Child Development at age five: a UK-US Comparison
Referee report
This manuscript describes associations between changes in neighborhood disadvantage and children’s cognitive and behavioral outcomes between ages one and five years in both the UK and the US. The paper is well written and interesting. I have some questions and suggestions that I hope the authors find helpful.
- Abstract –I find the wording of the third and second to last sentences of the abstract confusing. Also, the abstract leads the reader first to think neighborhoods do matter (“…attenuated but not eliminated by family circumstances”), but then concludes by saying that the family matters more than the neighborhood social profile. The conclusions of the paper seem to support the former. I suggest clarifying.
- Literature – My primary suggestion is to make the literature review more robust, especially by reviewing existing evidence on neighborhood effects (instead of simply the theoretical reason they might matter). The authors might consider including especially influential works in this area, such as those by Raj Chetty and others investigating the Moving to Opportunities Study. Related: the authors explain how their paper is a contribution, but the reader is left to take their word for it. A more robust discussion of related papers, and what they are missing that this paper fills in, would give the reader a better sense of how to situate this contribution into the broader body of knowledge.
- Methods -- I’m a bit confused about how the authors chose to handle children who did not move, but who lived in a neighborhood that changed significantly over this period of time (perhaps due to gentrification, economic investment, etc.). It seems that their change in area social advantage is coded as a zero, even if their neighborhood context changed around them. But then, why include them at all? Is the analysis identified only on children who move? It seems possible that moving itself (especially multiple times) is mechanism distinct from neighborhood change.
- Do the authors account for anything that happened to the children between ages 1 and 5? One thing I’m specifically wondering about is children who moved multiples times versus just one time, which is likely to predict outcomes and be different across countries.
- Generally, the authors are careful about avoiding causal language. However, I’d encourage one more close reading to improve even further here, avoiding words such as “influence”.
- Discussion -- The authors emphasize that their contribution is a cross-country comparison. However, I don’t believe they do an adequate job of explaining why this comparison is important, especially in the results section. What is the benefit of this comparison in terms of contribution to the literature? What does the comparison highlight besides similarities and differences across countries? What does describing these differences get us, given that the contexts are so very different?
